# Open Questions in Cold Atmospheric Plasma Treatment in Head and Neck Cancer: A Systematic Review

**DOI:** 10.3390/ijms231810238

**Published:** 2022-09-06

**Authors:** Vittoria Perrotti, Vito Carlo Alberto Caponio, Lorenzo Lo Muzio, Eun Ha Choi, Maria Carmela Di Marcantonio, Mariangela Mazzone, Nagendra Kumar Kaushik, Gabriella Mincione

**Affiliations:** 1Department of Medical, Oral and Biotechnological Sciences, University “G. d’Annunzio” Chieti-Pescara, 66100 Chieti, Italy; 2Department of Clinical and Experimental Medicine, University of Foggia, 71122 Foggia, Italy; 3Plasma Bioscience Research Center, Department of Electrical and Biological Physics, Kwangwoon University, Seoul 01897, Korea; 4Department of Innovative Technologies in Medicine & Dentistry, University “G. d’Annunzio” Chieti-Pescara, 66100 Chieti, Italy

**Keywords:** apoptosis, cold atmospheric plasma, head and neck cancer, RONS

## Abstract

Over the past decade, we witnessed a promising application of cold atmospheric plasma (CAP) in cancer therapy. The aim of this systematic review was to provide an exhaustive state of the art of CAP employed for the treatment of head and neck cancer (HNC), a tumor whose late diagnosis, local recurrence, distant metastases, and treatment failure are the main causes of patients’ death. Specifically, the characteristics and settings of the CAP devices and the in vitro and in vivo treatment protocols were summarized to meet the urgent need for standardization. Its molecular mechanisms of action, as well as the successes and pitfalls of current CAP applications in HNC, were discussed. Finally, the interesting emerging preclinical hypotheses that warrant further clinical investigation have risen. A total of 24 studies were included. Most studies used a plasma jet device (54.2%). Argon resulted as the mostly employed working gas (33.32%). Direct and indirect plasma application was reported in 87.5% and 20.8% of studies, respectively. In vitro investigations were 79.17%, most of them concerned with direct treatment (78.94%). Only eight (33.32%) in vivo studies were found; three were conducted in mice, and five on human beings. CAP showed pro-apoptotic effects more efficiently in tumor cells than in normal cells by altering redox balance in a way that oxidative distress leads to cell death. In preclinical studies, it exhibited efficacy and tolerability. Results from this systematic review pointed out the current limitations of translational application of CAP in the urge of standardization of the current protocols while highlighting promising effects as supporting treatment in HNC.

## 1. Introduction

Head and Neck cancer (HNC) is the sixth most common type of cancer by incidence worldwide [1], with an approximate 5-year survival rate of 65% [2,3], dropping to an average of 40% for patients diagnosed in advanced stage [4]; this is due to a poor response to standard anti-cancer treatments such as chemotherapy and radiotherapy [5,6], which has remained almost unchanged over the past decade [7]. The tumor is remarkably heterogeneous, based on localization in the oral cavity, oropharynx, larynx, and hypopharynx, on the types of cells within the tumor tissue and the molecular subtypes. The prognosis for patients with HNC is determined by the stage of the tumor at presentation, as well as the presence of lymph-node metastases and distant metastases [8]. Unfortunately, only one-third of patients were present with early-stage disease, whilst two-thirds were diagnosed with advanced cancer with lymph node metastases [5]. Surgery combined with adjuvant radiation therapy and/or chemotherapy is the standard of care, but in these cases, patients often experience notable complications related to disease treatments, including different degrees of hoarseness or aphonia, dysphagia, dry mouth, aspiration, neck masses, and facial distortion after surgery [9]. Indeed, it is important to obtain tumor-free resection margins in patients with HNC [10], and to achieve this, surgeons usually remove the tumor with a margin of 10 mm of macroscopically normal tissue, thus creating a large surgical defect that needs free flap reconstruction in more than half of patients [11]. Moreover, adjuvant chemoradiation to control the local disease itself may induce many morbidities that affect the quality of life and prognosis [12], led by the development of different side effects, such as permanent dry mouth, burning mouth syndrome, osteoradionecrosis, dental caries, neurological damage, thyroid function impairment, trismus, fibrosis and eye, and skin damage, compromising the patients’ recovery, psychological health, and physiology [13,14].

This scenario calls for an urgent need for novel methods in HNC treatment, and cold plasma-based systems may be a promising tool. The use of plasma sources in the medical field was approved in Germany in 2013, based on the results obtained regarding its effectiveness in stimulating skin regeneration and its antimicrobial effects [15]. Over the past years, plasma medicine has rapidly evolved due to the contribution of different groups of researchers in the development of medical devices generating cold atmospheric plasma (CAP) or non-thermal plasma (NTP) with temperatures at close to ambient temperature, thus allowing applications on heat-sensitive biological matters such as tissue [16]. Only in 2018 did it achieve an important milestone in the field of evidence-based medicine (EBM), reaching EBM level III in HNC [15]. Plasma is known as the fourth state of matter [17] and is partially or completely ionized gas that reacts with the environment creating a mixture of active components: electrons, ions, radicals, and energetic photons, all of which can, in turn, interact with the target [18]. It is a multicomponent system, a multiagent technology, and a multimodal therapy, and due to its unique traits, it elicits electromagnetic, chemical, and thermal effects on the target [19]. Due to advances in physics and biotechnology, nowadays, plasma can be easily generated and used at room temperature reaching high electron temperatures, but very low gas temperatures are associated with weak ionization rates [20]. 

CAP devices have been designed with different configurations [21]: dielectric barrier discharge (DBD), corona discharge, radio frequency reactor, or direct current plasma jet with different carrier gas [6]. In that respect, most CAP sources are operated in the noble gasses’ environment, e.g., helium or argon, which are characterized by a relatively low breakdown voltage and long plasma sustainability [22]. They are commonly mixed with a small number of other gasses (e.g., O_2_) to increase their chemical reactivity [23]. Recently, CAP has been investigated in several medical applications, such as blood coagulation [24], wound disinfection and healing [25], tissue and germicidal irradiation and sterilization [26], skin rejuvenation [27], tooth bleaching [23], material surface modifications and crosslinking [28,29,30], and to selectively eradicate cancer cells [31,32,33]. In this last specific aim, two different types of tumor treatment using CAP have been described: direct exposure to a plasma jet or indirect exposure by a medium (named plasma-activated medium, PAM) previously exposed to cold plasma (Figure 1) [34]. Both types of treatment decrease cell viability with the benefit that indirect treatment using PAM made CAP treatment an innovative drug therapy [35].

The anti-cancer effect of CAP is a combination of physical and chemical factors, and it depends on treatment time, working gas, treatment pattern (direct or indirect), and cell/tissue type. Ultraviolet, heat, and electromagnetic fields are physical factors in CAP. Chemical factors include reactive species produced in the gas phase of CAP. Among them, oxygen-reactive species such as hydroxyl (OH), superoxide (O_2_^−^), hydrogen peroxide (H_2_O_2_), as well as nitrogen-based species such as nitric oxide (NO), nitrogen dioxide (NO_2_), nitrogen trioxide (NO_3_), have been demonstrated. One key aspect, which is attracting increasing attention, is the ability of CAP to induce cell death in various cancers, including skin, liver, and colon cancer [36], without damaging the surrounding healthy tissues; however, the mechanisms remain poorly understood. The selectivity of plasma in inactivating tumor cells over non-malignant cells is a matter of debate, although it is generally accepted that reactive oxygen and nitrogen species (RONS) are the major agents responsible for plasma-promoted biological outcomes. Indeed, it has been shown that the physical effects seem to have a minor biological impact, while RONS may induce intracellular reactive oxygen species (ROS) increase, a decrease in antioxidant enzymes, DNA double-strand brakes, and consequently, apoptosis, according to cell type and exposure parameters (total delivered energy, power, time of exposure). Indeed, excessive ROS or reactive nitrogen species (RNS) induces oxidative stress against cancer cells and, by damaging essential cellular molecules, i.e., cell cycle checkpoints, deoxyribonucleic acid, and other organelles subsequently, results in apoptotic death. However, the mechanisms of CAP action are not well understood at a molecular level. Since the first report about the killing effect of a plasma jet on HNC in 2012 [37], the field of CAP application in this type of cancer treatment has expanded rapidly. 

In metastatic or recurrent HNC therapy, resistance and multiple severe side effects of conventional anti-cancer therapies such as chemotherapy limit the benefit of single-agent therapies. Indeed, numerous clinical evidence indicates that single-agent systemic therapies infrequently culminate in long-lasting medical responses. Factors such as tumor plasticity, responsible for therapeutic adaptation, and heterogeneity, which induce the emergence of therapy-resistant tumor clones, can cause the failure of single-agent therapy. To overcome such phenomena, CAP might exhibit an important adjuvant treatment effect. Lee et al., 2020 [38] analyzed the synergistic effect of the combination of CAP and cisplatin-mediated chemotherapy and found that ROS generation, tumor-suppressor proteins, and apoptosis-related enzymes increased according to the treatment time of CAP in combination with cisplatin, thus showing the synergistic effect of cisplatin and CAP treatment against SCC-15 cells with low cytotoxicity against normal cells. Moreover, CAP can obtain a stronger anti-cancer capacity through the synergistic application with nanoparticle technologies, as demonstrated by Choi et al. [37]. Finally, the synergy found by Lafontaine et al. [39] between gas plasma plus radiation therapy and gas plasma combined with a chemotherapeutic agent provides evidence for further opportunities to improve cancer treatments. 

Currently, many research groups are concentrating their efforts on the cell-specific effects of CAP treatment, exploring the fundamental mechanisms at the molecular level and the optimal parameters for cancer cell treatment, with the long-term goal of using plasma as an alternative means of effective cancer therapy. The purpose of the present article was to conduct a systematic review of the current evidence, focusing on CAP as potential therapeutics in HNC, to summarize the characteristics and settings of the different devices to meet the urgent need to standardize protocols, procedures, and multidisciplinary guidelines and to evaluate and compare the effectiveness of CAP treatments in a laboratory setting, with possibilities to extrapolate these methods to clinical tests.

## 2. Materials and Methods

This systematic review was performed according to the guidelines of the Preferred Reporting Items for Systematic Reviews and Meta-analyses (PRISMA) statement [40].

### 2.1. Search Strategy

An electronic search on scientific databases (PubMed, Scopus, Web of Science) was performed to identify suitable studies, using the following terms and keywords alone or in combination: (cold atmospheric plasma OR cold plasma OR low temperature plasma OR kinpen med OR non-thermal atmospheric pressure plasma OR non thermal atmospheric pressure plasma OR cold physical plasma OR plasma medicine OR CAP OR plasma activated medium OR cold atmospheric-pressure plasma OR plasma activated liquid OR cold argon plasma) AND (head and neck tumors OR head and neck squamous cell carcinoma OR head and neck cancer OR head and neck cancer OR oral squamous cell carcinoma OR oral cancer OR oral squamous cell cancer OR OSCC OR oral oncology OR oral squamous cancer cell OR oral cavity squamous cell carcinoma) AND (therapy* or treat* OR application or attenuation OR efficacy OR action OR effect* OR target* OR response).

The first search was performed on 13 December 2021. The search was limited to studies published in the English language. The last electronic search was performed on 10 August 2022. In addition to the electronic search, a hand search was undertaken by checking the references of the included studies to identify further eligible studies. A reference manager software program (Endnote X9.3.2, Clarivate Analytics) was used, and the duplicates were discarded first electronically, then by checking the resulting list manually. Search strategy for PubMed, Web of Science, and Scopus is shown in Appendix A. 

### 2.2. Eligibility Criteria

#### 2.2.1. Inclusion Criteria

Clinical prospective studies with a cross-sectional, cohort, case-control, and case-series or report design were included as well as preclinical studies on both in vitro and animal models. Patients included in the study had to be over 18 years old and be diagnosed with HNC [41], and treated with CAP. Biological, physic-chemical, and/or clinical outcomes had to be reported, such as efficacy and safety. Specifically, for preclinical studies, in vitro and in vivo assays had to be performed on HNC cell lines or xenografts derived from HNC cell lines or patient-derived xenografts (PDX).

#### 2.2.2. Exclusion Criteria

Systematic review, meta-analyses, and ecologic studies.

### 2.3. Focused PICO Question

Participants: patients with diagnosed HNC as well as HNC cell lines or xenograft models.

Intervention: documented protocol of direct or indirect CAP irradiation.

Comparison: the studied model (cell line, xenograft, or human) exposed to other kinds of treatment or different CAP treatment protocols. 

Outcomes: primary outcomes were cell viability and apoptosis, while secondary outcomes included protein expression and ROS production in vitro studies and in vivo efficacy in animal and human studies.

### 2.4. Selection of Studies

Retrieved citations were independently screened by two authors (VP and VCAC), and relevant studies were identified based on title and abstract. If those did not provide sufficient information about the inclusion criteria, the full text was evaluated to assess eligibility. Any disagreement was solved by discussion, and a third reviewer was consulted to make final decisions (MM). This author also calculated a k-statistic value to ascertain the level of reviewers’ agreement.

### 2.5. Data Extraction and Method of Analysis

Two reviewers (VCAC and MCDM) independently extracted data from all the included studies using a predesigned extraction form. Microsoft Excel 2020 (Microsoft Office, Microsoft Corporation, Redmond, WA, USA) was used for data collection and for descriptive analysis. 

The following data were collected: type of plasma device and trade name if present, CAP settings (i.e., Flow rate, Gas, pulse frequency, pulse, temperature, application distance, application time, total energy, and power), type of cancer target, study level, study population/model, tumor stage, type of treatment (i.e., direct, or indirect), sample size, quantitative measures, developer, references, and notes. The primary outcomes included safety and efficacy of CAP in clinical studies. For preclinical studies, cell viability, DNA damage, changes in cell morphology, and apoptosis after CAP exposure among groups or difference in tumor dimension means in xenograft models were the main investigated outcomes. 

### 2.6. Risk of Bias Assessment

Two reviewers (VP, VCAC) assessed the quality of studies by risk of bias evaluation. Because of the different study designs of all the included studies, diverse methods of assessment were used. For in vitro studies, no standard quality assessment tool exists [42]; therefore, we developed these criteria by ourselves. For the methodological quality of in vitro studies, criteria based on the parameters for developing cell culture studies were adapted [43]. In particular, 6 items were evaluated as adequate or inadequate, whether information about: 1. condition of cell culture before treatment; 2. condition of cell culture during the treatment; 3. description of methodology to evaluate outcomes; 4. case-control description; 5. multiple experiments performed; 6. descriptions of plasma devices and settings were reported. Studies fulfilling at least five over six of the above-reported items were considered at low risk of bias, while two or more than three inadequate reported items were classified respectively at medium and high risk of bias. Regarding in vivo animal studies, the Systematic Review Centre for Laboratory Animal Experimentation’s (SYRCLE) risk of bias tool was used to assess the quality of available evidence [44]. At last, for human clinical studies, Newcastle-Ottawa Scale was used as a qualitative tool (NOS) in case of cohort or case-control study design [45], while case-series studies were evaluated by Murad’s quality checklist of case series and case report [46], without considering items 5 and 6 since they are more relevant of drug investigational studies [47].

### 2.7. Statistical Analysis

Considerable methodological and clinical heterogeneity was found in the selected studies regarding participants, interventions, and outcome measures. Evident differences were seen in study design, methodologies, treatment modalities, outcome measures, and results. Moreover, many studies reported aggregated results with only descriptive or graphic representation of CAP-related outcomes and without a standardized system. Finally, not all the studies performed statistical analysis of data; therefore, comparisons between studies were not feasible, and a descriptive presentation had to be adopted since meta-analysis was considered inappropriate. However, when possible, quantitative data were presented. Moreover, when applicable, based on each study categorization and criteria, descriptive results were arranged ordinally, and comparisons were made within each factor.

## 3. Results

### 3.1. Study Selection

A total of 9413 potentially relevant records were identified from the databases and further processed following the PRISMA statement (Figure 2). 

After the removal of duplicates, 1997 records were further examined based on their titles and abstracts, and 1950 studies were excluded as they did not meet the specific eligibility criteria for this study. A total of 47 full-text articles were finally evaluated, and 23 were subsequently excluded [48,49,50,51,52,53,54,55,56,57,58,59,60,61,62,63,64,65,66,67,68,69,70] for the reasons listed in Table 1. A total of 24 studies [5,6,15,23,36,37,38,71,72,73,74,75,76,77,78,79,80,81,82,83,84,85,86,87] fulfilled the selection criteria of the present review and were included for qualitative analysis. The value of the k-statistic was 0.78, which indicates an excellent level of agreement between reviewers

### 3.2. General Characteristics of Included Studies

Based on the development status, the studies were categorized into two main groups: 


-Nineteen [5,6,23,36,37,38,71,72,73,74,75,76,77,78,79,80,81,82,83] preclinical studies investigated the CAP in vitro effect on different HNC cell lines, and 3 [73,74,77] out of those 19 articles conducted in vivo experiments on an animal model (Table 2);-Five [15,84,85,86,87] clinical studies were conducted on human beings (Table 3).


### 3.3. Results of Risk of Bias Assessment

A low risk of bias was found for all the in vitro studies. Only in one study [23] was there inadequate information about the “condition of cell culture before treatment” domain, while three other studies lacked “multiple experiments performed” evidence. All three in vivo animal studies [73,74,77] were at high risk of bias based on SYRCLE features. Only three domains, such as “sequence generation”, “baseline characteristics”, and “selective outcome reporting”, were adequate in all the studies. “Blinding” in outcome assessment was adequate only in Kang et al. [73], while “incomplete outcome data” was adequate only in the Oh et al. study [77]. Information about the following domains, such as “allocation concealment”, “random housing”, “blinding”, “random outcome assessment”, and “other sources of bias”, were not retrievable. On the other hand, NOS for case-control studies resulted in adequate in all the items for the only study included by Dai et al. [84], while Schuster et al. (2016) [87] was inadequate in two items, such as “representativeness of exposed cohort” and “selection of non-exposed cohort”, based on NOS for cohort studies. At last, for case series studies, based on Murad et al. [46] risk of bias assessment, all three studies were at low risk of bias. Metelmann et al. (2018) [15] was adequate in all the evaluated items, while Schuster et al. [87] and Metelmann et al. (2015) [85] were inadequate in the “selection” process. Detailed risk of bias assessment and evaluation is reported in Appendix A. The relevant studies were conducted in various centers around the world, mainly in Europe and Asia, and were published between 2012 and 2021 (Figure 3).

### 3.4. Preclinical Model: Cell Lines

The 19 in vitro studies [5,6,23,36,37,38,71,72,73,74,75,76,77,78,79,80,81,82,83] evaluated the CAP effect on various HNC cell lines (Figure 4) arising at specific anatomical sites and representative of specific cancers, such as the human floor of the mouth squamous cell carcinoma (AMC-HN6, Ho-1-u-1), human tongue squamous cell carcinoma (CAL-27, HNO-97, HSC-3, HSC-4, OSC-19, SAS, SCC-15, SCC-25, SCC-4), human oral cavity squamous cell carcinoma (HSC-2, SCC-QLL1), human oral retromolar trigone cavity squamous cell carcinoma (SCC-1483), human hypopharynx squamous cell carcinoma (FaDu, SNU-1041), human laryngeal squamous cell carcinoma (JHU-022, JHU-029, SNU-899, SNU-1076), human pharyngeal squamous cell carcinoma (Detroit-562), human undifferentiated carcinoma of the parotid gland (AMC-HN-9), and normal oral cavity cell lines: human gingival fibroblasts (HGF-1), human normal oral keratinocytes (NOKsi) and normal human floor of the mouth keratinocytes (OKF6/T). 

Fourteen studies [5,23,36,37,38,71,73,75,78,79,80,82,83] evaluated only the effects of direct CAP application, while four studies [6,72,74,77] applied indirect CAP exposition; only two studies in in vitro model [81] compared both effects of direct and indirect plasma exposition. Different in vitro tests were performed to assess biological and chemical reactions. Fifteen studies [5,6,37,38,72,74,75,76,77,78,79,80,81,82,83] evaluated cell viability, twelve studies [5,36,37,38,72,73,74,75,77,78,79,80] investigated protein expression changes, twelve studies [36,38,72,73,74,75,77,78,80,81,82,83] focused on apoptosis, four studies [23,36,71,82] evaluated the potential of plasma to determine DNA-damage, three studies [36,37,77] observed changes in cell morphology, another three [5,74,80] performed colony formation assay, and finally four studies [36,72,75,79] investigated changes in cell cycle. Two studies furtherly quantified changes in gene expression [73,76], two assessed the presence of mitochondrial damage [73,77], and two assessed changes in cell motility by migration and invasion assays [72,80]. Only one study investigated changes in catalytic Fe (II) [80]. Eleven studies [23,38,73,74,75,76,77,79,80,81,83] assessed the chemical characteristics produced by plasma exposition by quantification of reactive-species production (Figure 5A,B). 

***Cell viability assays:*** Seven studies [5,38,74,79,81,82,83] performed MTT or MST assay, while two studies [37,76] investigated cell viability by trypan blue staining. Park et al. [78] performed SRB-assay; Chauvin et al. [6] PrestoBlue^®^ assay, Hasse et al. [72] performed Resazurin-based assay, while Sato et al. [80] and Oh et al. [77], respectively, used WST-8 and WST-1 assays. Sklias et al. [81] performed MTT, trypan blue and CellTiterGlo^®^ assays. In general, a higher reduction in cell viability was observed in cancer cell lines compared to normal ones, with variability among the diverse protocols, but an overall increase in cancer cell deaths was observed in a dose-time-dependent manner. Better results were also obtained when combining plasma treatment with GNP-EGFR, PD-L1-ab, and GNP or cisplatin. Three studies [5,74,80] furtherly assessed cell survival based on the ability of a single cell to grow into a colony.

***Protein expression:*** TP53 gene-related pathway was mainly investigated by Western-Blot analysis, quantifying different proteins. One study [36] investigated the p-ATM, p-53, p21, cyclin D1mm, and yH2AX axis. Another study [79] investigated p-ATM—Bcl-2 changes. PARP—caspase 3/7—Bax/Bcl-2 axis was investigated in two studies [77,78], while one study [5] investigated PARP expression alone. PTEN-p53-Cas-9 axis was also investigated in one study [71]. Other evaluated proteins were p-p38/p-c-JUN/JNK/p-ERK axis, AKTmm, and MMP-2; one study [75] quantified EGFR changes. 

***Apoptosis:*** Four studies evaluated apoptosis by FACS/TUNEL assays [36,65,80,83], while four and two studies respectively performed only FACS [38,75,77,82] or TUNEL assay [72,74], respectively. Park et al. [78] evaluated apoptosis by immunocytochemistry and by quantification of cleaved caspase-3, PARPm, and cytochrome C. Sklias et al. [81] analyzed the Caspase 3/7 signal; an overall increase in apoptotic mechanisms was observed in cancer cells compared to controls. In particular, Kang et al. [73] highlighted how direct plasma treatment led to apoptosis by MAPK-mediated mitochondrial ROS. 

***DNA-damage:*** Two studies [23,71] performed immunofluorescence assays, and two studies [36,82] performed COMET assay. Overall, an increase in DNA damage/fragmentation was observed in a time-dose-dependent manner. Of interest, Han et al. (2020) [71] observed maximized DNA damage in the plasma treatment region, which decreased radially outward. Moreover, S-phase cells were more susceptible than G_1_-G_2_-phase cells. 

***Cell morphology:*** SEM was employed in two studies [37,77], while light microscopy was used in one study [36]. Overall, changes in shape (from spindle to round), cell detachment, shrinkage, and membrane ruptures were observed. 

***Cell cycle:*** All four studies [36,72,75,79] investigating changes in cell cycles employed FACS assay. Chang et al. [36] reported that CAP caused cell cycle arrest in the sub-G_1_ phase, and the proportion of SCC-25 cells (expressing wild-type p53) in the sub-G_1_ phase increased markedly in an intensity-dependent manner. However, the other cell lines (MSK QLL1, SCC-1483, and SCC-15) with mutated p53 were unaffected by CAP treatment. Similar results were obtained in Lee et al. (2016) [75], where HSC-2 cells were killed via sub-G_1_ phase arrest, and in Ramireddy et al. [79], where CAP-induced cell death and G_1_ cell cycle arrest associated with the ATM/P53 pathway.

***Gene expression, mitochondrial damage, and cell motility:*** Kim et al. (2015) [74] performed qRT-PCR analysis showing that plasma treatment led to MUL1 mRNA expression, with increased cytotoxicity by MUL1/AKT binding. Oh et al. [77] also performed qRT-PCR analysis, investigating the role of ATF4 and CHOP as key regulators in non-thermal atmospheric plasma-activated media (NTPAM) induced cell death. Mitochondrial damage was assessed in the studies of Kang et al. [73] and Oh et al. [77]. The first study was performed by FACS and MMP measurements [73], while the second one conducted OCR assay and SEM [77]. In both studies, there was evidence of mitochondrial damage and dysfunction. Hasse et al. [72] and Sato et al. [80] investigated cell motility by wound healing assay, showing a marked reduction in cell motility of treated cancer cells without or with irrelevant changes in normal cells. 

***Chemical characteristics:*** Strong evidence emerged suggesting the development of reactive species after plasma treatment. Three studies [76,79,85] quantified the formation of such reactive species by fluorimetric assay, three studies by OES [38,75,81], three studies [36,74,80] by FACS, and one study [83] by OxiSelect Intracellular assay. Based on plasma device and substrate, most reported species were NO-γ, O^2−^, more general ROS, thiol, Ar^+^, OH^−^, RNS, H_2_O_2_, NO_2_^−^ and NO_3_^−^.

### 3.5. Preclinical Models: In Vivo Animal Studies and Ex Vivo Human Studies

Only four studies [73,74,76,77] applied plasma treatment to preclinical models. In Kang et al. [73] study, the plasma treatment was delivered directly to FaDu-derived xenograft models in mice. By immunohistochemistry (IHC) It was observed an increase in caspase-3 and Nox-3 levels and an inhibition of tumor growth after 11 days by tumor mass measurement by a sliding caliper. Kim et al. (2015) [74], instead, employed both an SCC7-derived syngeneic tumor model and an SCC15-derived xenograft in mice. In this case, plasma treatment was not administered directly but by injection of a liquid-type NTP (LTP). Similarly, an inhibitory effect on tumor growth and development was observed. Furtherly, by IHC and Western blot assay, p-AKT was reduced while MUL1 increased. Oh et al. [77] performed a similar study by injection of NTPAM in an SNU-1041-derived xenograft model in mice. After NTPAM treatment by IHC, they observed low expression of Ki-67 and high expression of ATF4 and CHOP. Lower tumor weight was also observed in the test group compared to the control. At last, Hasse et al. [72] directly targeted human tissues. Their study showed an increase in cytochrome C, INF-γ, TNF-α, and IL-10 levels, while IL-22 expression increased in healthy tissue. 

### 3.6. Clinical Model

Five studies [15,84,85,86,87] investigated the consequences of plasma treatment on human clinical models. A total of 159 patients were included in different kinds of study setups. Fifty-three patients were diagnosed with HNC, six with oropharyngeal squamous cell carcinoma (OPSCC), and 100 with laryngeal carcinoma (LC). In four studies [15,85,86,87], Ar-CAP kINPen MED was used directly on ulceration, vertically to the naturally moist tissue surface. In one study [84], they employed Unitec low-temperature plasma operation system, directly ablating the tumor up to 3-5 mm away from the edge of the lesion. Outcomes varied among studies to address the effects of plasma on contamination of infected ulcerations, side effects, pain, tumor surface changes, and effects on tumor growth. Overall, the plasma treatment resulted in a few patients experiencing mild–moderate side effects and was never life-threatening. Of interest, a significant reduction was observed in terms of fetid odor, contamination, and pain of lesions (Table 3). 

### 3.7. Types of CAP Devices and CAP Treatment Parameters

In the 24 included studies, plasma treatment was delivered directly by 13 different kinds of devices, and kINPen MED was employed in four studies [15,85,86,87], while a PAM produced by the same device was delivered in one study [72]. He + O_2_ was used in two studies [36,73] directly, while two studies [74,77] exposed cells to an LTP produced by this device; only in Sklias et al. [81] was the mixture used both directly and indirectly. Similarly, He-CAP was used in two studies [5,79] directly, while another study [6] served as a PAM produced by the He-CAP jet. The most used gas was Ar in eight studies [15,38,72,78,80,85,86,87], followed by He + O_2_ [36,65,66,69,73] and finally He alone [5,6,79], N_2_ alone [23,71,75] and air [37,76,82] in three studies each; He + N_2_ was delivered only in one study [83] (Figure 6). The study of Dai et al. [84] lacked this information. Application protocol widely varied among studies, including pulse frequency, pulse, flow rate, plasma temperature, source application distance and time, total emitted energy, and power. Overall, the pulse ranged between 1.2 and 30 kV; pulse frequency ranged between 0.05 and 30 kHz, while two studies [75,80] reported 60 Hz, and in the study, a pulse frequency of 13.56 MHz was applied [83]. While application distance and time ranged between 2 and 30 mm and 1 and 300 s, respectively, most of the studies lacked information about temperature, energy, and power. In humans, a pulse frequency of 1 MHz was mainly used. There is still an incomplete understanding of the critical CAP treatment characteristics for effective therapeutic clinical dosing. All the parameters used in the included 24 studies are summarized and outlined below (Table 4).

## 4. Discussion

The anti-cancer mechanism of CAP treatment is still an open question. In this systematic review, a great variability regarding CAP conditions of use has emerged, and this calls for an urgent need for protocols’ standardization to evaluate and compare the effectiveness of plasma treatments in a laboratory setting, with possibilities to extrapolate these methods to clinical tests and finally to favor the clinical translation of plasma as a precision medicinal approach. In contrast to traditional anti-cancer treatments, the primary benefit of CAP is its selectivity toward tumor cells which is an essential feature in the era of targeted therapies [89]. Indeed, the therapy of tumors with anti-cancer drugs faces three important problems: specificity of the treatment, resistance of tumor cells, and penetration of the treatment [90]. Due to its physical—ultraviolet, heat, and electromagnetic field—and chemical—reactive species produced in the gas phase of CAP—properties, CAP is a multimodal therapeutic tool [91] that could offer an answer to each of these challenges and provide further opportunities to overcome critical issues such as tumor plasticity and heterogeneity, which are accounted for the causes of the failure of traditional approaches [82,92].

Understanding CAP selectivity is one of the key challenges in the field of plasma cancer therapy. Several hypotheses have been advanced to explain the selectivity of CAP; however, two models have been proposed [93]. First, further oxidative stress can more easily exceed the cellular survival limit in cancer cells. Indeed, due to stronger metabolism in cancer cells, the basal ROS level in these cells is higher than that in normal cells [50,94,95]. When supplementary ROS stress is generated by CAP treatment, the intracellular ROS level in cancer cells overcomes a threshold more easily than that in normal cells [96], resulting in the activation of apoptotic cell death pathways. However, the basal-ROS level model can just explain the production of higher ROS levels in the CAP-treated tumor cells but cannot explain the selective increase in ROS only in cancer cells. Nevertheless, it is generally accepted that RONS are the major agents responsible for CAP-promoted biological outcomes [97]. Indeed, it has been shown that the physical effects seem to have a minor biological impact, while RONS may induce intracellular ROS increase, antioxidant enzymes decrease, DNA double-strand brakes, and consequently, apoptosis, according to cell type and exposure parameters [98]. Ramireddy et al. [79] showed that H_2_O_2_ generation increased immediately after He plasma treatment accounting for the anti-cancer effects observed in OSCC cell lines. Lin A. et al. [76] demonstrated that the generation of RONS and cell death depended on the total delivered energy during treatment, while it is independent of pulse frequencies and treatment times. Both direct and indirect CAP applications produce different RONS environments, although there are no remarkable distinctions in their effects [99]. Indeed, both types of treatment decreased HNC cell viability by triggering more apoptosis in cancer cells than in normal cells, with the benefit that indirect treatment using PAM makes CAP treatment an innovative drug therapy [6]. To note, Oh et al. [77] showed that PAM reduced HNC cell viability in a time-dependent manner while cytotoxicity was not observed in the normal control cell line. In addition, annexin V-FITC/PI analysis showed statistically significant enhancements of the fractions of cells corresponding to early apoptotic cells (FITC+/PI−) and late apoptotic cells (FITC+/PI+) [77]. The decreased viability of OSCC vs. HGF cell lines was also shown by Lee et al. [75], who demonstrated the absence of changes in cell cycle components in normal cells after CAP treatment. Moreover, they showed that CAP led to degradation and dephosphorylation of EGFR, overexpressed in OSCC cells, through a mechanism NO-dependent as revealed by pretreatment of cells with NO scavenger [75]. However, the selectivity was not always established. Indeed, in the study by Hasse et al. [72], normal keratinocytes and HNC cell lines revealed very little difference with respect to the induction of apoptotic events after CAP exposure, although these results were in contrast with the ex vivo experiments, which demonstrated a predominant apoptosis induction in tumor tissue while in healthy tissue the impact of CAP was not significant. Moreover, Sklias et al. [81] demonstrated that indirect plasma treatment is as efficient at killing tumor cells as an appropriate combination of H_2_O_2_, NO_2_^−^ and acidic pH in ad hoc solutions, while sparing normal cells.

Apoptosis was the most investigated pathway in the included studies. Lee et al. [75] showed an increase in apoptosis detected in HNC cells killed via sub G_1_ arrest through Annexin V-PI staining. He-CAP induced cell death and cell cycle arrest and activated mitochondria-mediated apoptosis by enhancing Bax expression and suppressing Bcl-2 protein expression by increasing intracellular RONS [79]. Hasse et al. [72] showed a cell death mechanism based on caspase 3/7 pathways activation. The main pathway leading to apoptosis was triggered via DNA and mitochondrial damage. Oh et al. [77] observed reduced viability after NTPAM treatment of three HNC cell lines due to enhanced apoptosis associated with an increase in mitochondrial ROS by upregulation of ATF4, a transcription factor expressed in the context of sustained endoplasmic reticulum and mitochondrial stress, and CHOP, a molecule activated by ATF4 that regulates the expression of pro-apoptosis-related genes. According to Kim SY et al. [74], plasma-induced apoptosis was linked to AKT1 ubiquitination and degradation initiated by the mitochondrial protein MUL1, an E3 ligase known to regulate cell growth and death. On the other hand, Chang et al. [36] showed the influence of p53 mutational status on CAP treatment and demonstrated that DNA damage was dependent on the ATM/p53 signaling pathway by sub-G1 arrest. According to Kang et al. [73], apoptosis was induced by MAPK-dependent mitochondrial ROS. Double strand breaks (DSB) are an important mechanism of DNA damage induced by CAP [23]. The number of cells with DSB varied due to the distance from the irradiation center and duration of exposure [23]. Only Chauvin et al. [6] compared the response to PAM treatment in monolayer cultures (2D) and multi-cellular tumor spheroids (MCTS) and found that multiple treatments were needed to obtain a total disruption of spheroids, demonstrating that MCTS models mimic closer to an in vivo tumor behavior. By contrast, in some cases, the involvement of a viability inhibition mechanism mediated by apoptosis has not been demonstrated. Indeed, Guerrero-Preston et al. [5] found that CAP selectively diminished HNC cell viability in a dose-response manner; however, Western blot analysis did not provide evidence that the cleavage of PARP occurred following CAP treatment, thus suggesting that CAP selectively impairs HNC cell lines through non-apoptotic mechanisms while having a minimal effect on normal oral cavity epithelial cell lines. In Hasse et al. [72], ex vivo tissue biopsies treated with CAP revealed an increased number of apoptotic cells within the tissue paralleled by increased levels of cytochrome C in the extracellular liquid, suggesting apoptotic cell damage, in accordance with clinical findings in HNC patients [87].

Finally, the investigation of CAP combination treatments, although in their infancy, demonstrated promising results. Lee et al. [38] found a synergistic effect of the combination of CAP and cisplatin-mediated chemotherapy on ROS generation, apoptosis, and anti-cancer activity with low cytotoxicity against normal cells. Choi et al. [37] demonstrated a significant decrease in the viability of cells pretreated with an anti-EGFR antibody conjugated with gold nanoparticles before CAP irradiation. Finally, Park et al. [78] investigated a novel combination therapy by gold nanoparticles conjugated to programmed cell death protein ligand 1 (PD-L1) antibodies and CAP, which resulted in a greater number of dead cells compared to other experimental groups. 

## 5. Conclusions

In the last years, growing evidence supports the effectiveness of plasma-produced RONS by eliciting a broad spectrum of effects. Perhaps the most relevant concern is the variability in RONS generation due to variations in environmental or sample conditions. Welz et al. [82] emphasize that a comparison of various experiments carried out with different CAP devices is difficult due to different CAP parameters and designs, i.e., DBD (dielectric barrier discharges) technology, SMD technology, power input, voltage, frequency, carrier gas resulting in relevant differences in the production of their components. Moreover, experiments performed with the same plasma device also led to different results when diverse cell lines (malignant/non-malignant) were targeted. 

Thus, only delivering controllable doses of RONS to target biological systems can elicit specific effects such as cell death, inhibition of cell proliferation, and migration. Therefore, since any successful medical technology must be predictable and repeatable, the dose for medical application of plasma, the monitoring, and control are the challenges in dealing with the application of plasma in the medical field. Consequently, there is an urgent need for standardization of plasma sources, reliable protocols, and multidisciplinary guidelines to evaluate and compare the effectiveness of CAP treatments in a laboratory setting, with possibilities to extrapolate these methods to clinical tests. Last but not least, a dialogue between plasma experts (physicists, chemists, and engineers), biologists, and physicians to boost the information exchange and, therefore, the publication of results, is strongly required.

Moreover, to be considered an anti-cancer treatment approach with a solid theoretical basis, the underlying mechanisms driving CAP-induced biological effects need to be fully understood. Synergistic effects of CAP with chemotherapy and immunotherapy are demonstrated in vitro [38,78]. The aim of such combination treatments is to reduce effective doses of radiation or chemotherapy drugs, thus lowering the side effects. Traditional surgery and chemo- and radiotherapy may be associated with psychological effects in HNC patients due to potential negative changes in appearance as well as physiological implications of functional defects (speaking, eating, and even breathing) of affected facial organs. Such patients may benefit from CAP treatment to selectively clear the margin of cancer cells without the need to remove large areas of normal tissue. Using the antimicrobial effect induced by plasma, CAP application was successful in palliative care of patients with advanced HNC presenting infected tumor ulcerations. Because of the treatment, the microbial load and the resulting typical fetid odor were reduced. Apart from additional pain reduction, in some cases also, transient tumor remission occurred [15,85,87]. Therefore, CAP can also be used to treat the remaining healthy tissues to decontaminate the site, thus removing pathogenic microorganisms, promoting blood coagulation, and stimulating regeneration and healing of healthy tissue. This could significantly improve the quality of life of HNC-affected patients by decreasing the microbial burden on tumors and enhancing social interaction—an aspect that by itself should be a motivator to scientists in the field of plasma oncotherapy. However, tumor recurrence was observed. This result raises the urgency to understand why the anti-tumor effect does not persist over time and the need for clinical trials with longer follow-ups.

## Figures and Tables

**Figure 1 ijms-23-10238-f001:**
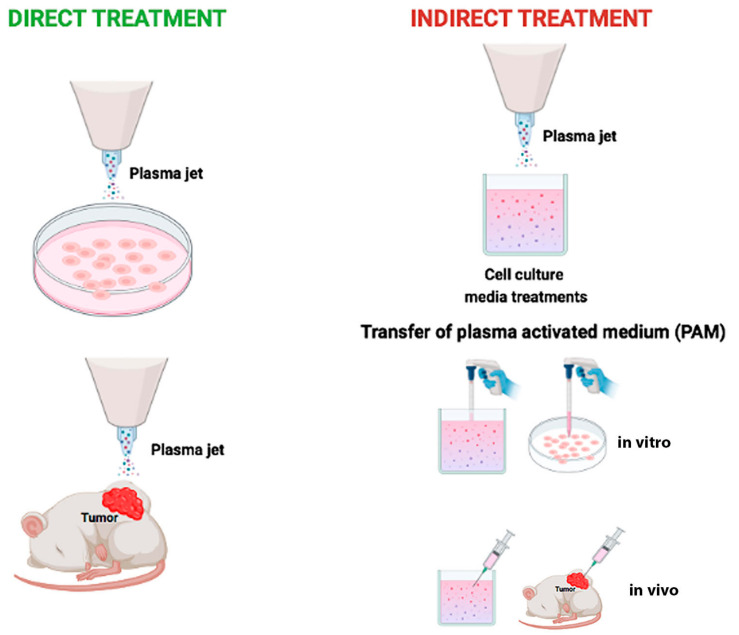
Schematic representation of two basic strategies to use cold atmospheric plasma. Direct cold atmospheric plasma treatment on cancer cells in vitro or on subcutaneous xenograft tumors in vivo (**left**). Indirect cold atmospheric plasma treatment on solutions, mostly medium. Plasma Activated Medium is used to treat cancer cells seeded in dish or tumor tissues in mice (**right**).

**Figure 2 ijms-23-10238-f002:**
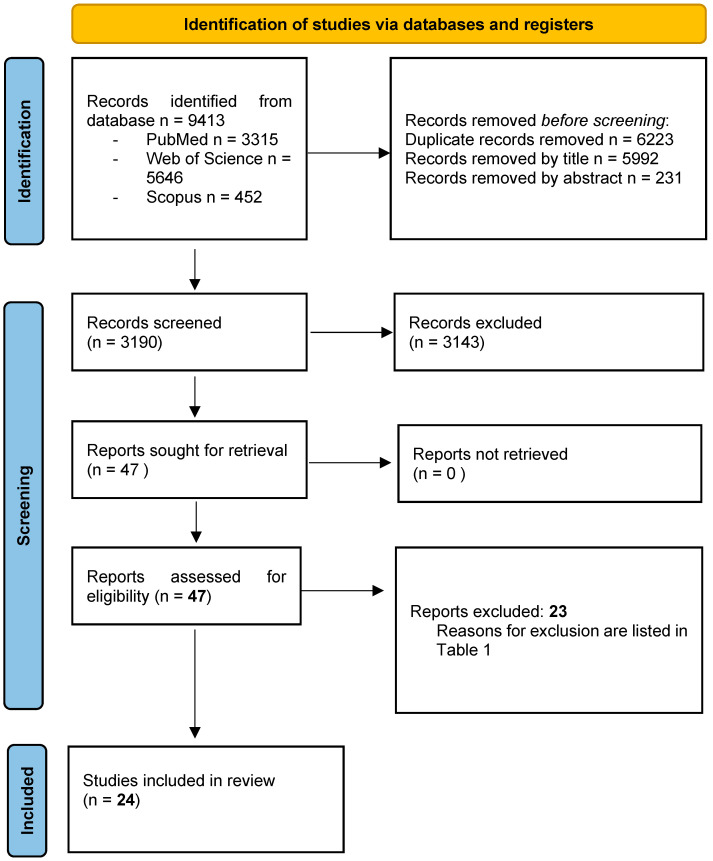
PRISMA 2020 Flow Diagram of the screening process. In total, 24 studies were included in the present systematic review.

**Figure 3 ijms-23-10238-f003:**
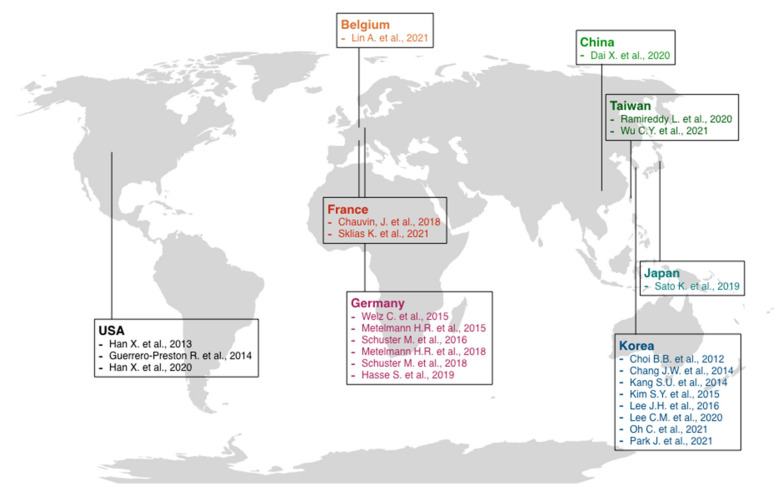
Schematic representation of the most relevant cold atmospheric plasma studies [5,6,15,23,36,37,38,71,72,73,74,75,76,77,78,79,80,81,82,83,84,85,86,87] published in various countries.

**Figure 4 ijms-23-10238-f004:**
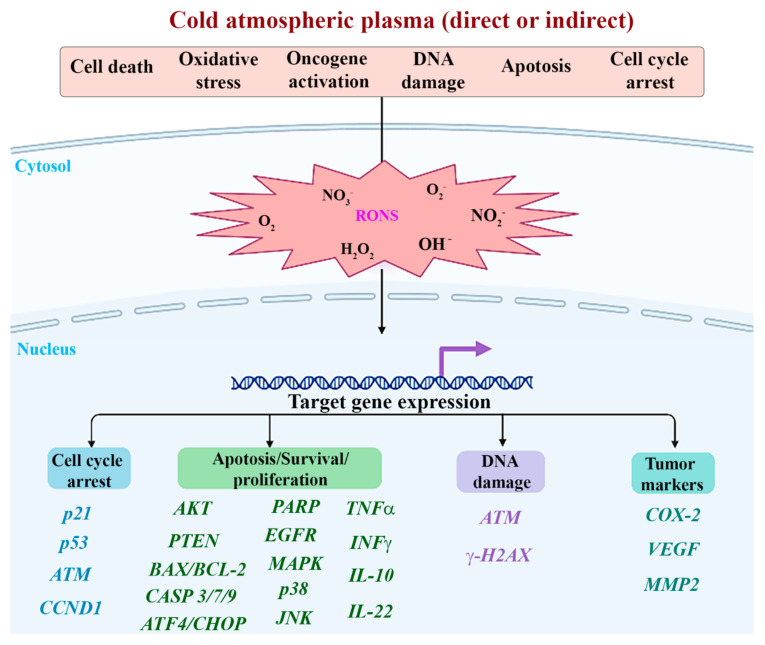
Molecular mechanisms induced by direct and indirect application of cold atmospheric plasma included in the present systematic review.

**Figure 5 ijms-23-10238-f005:**
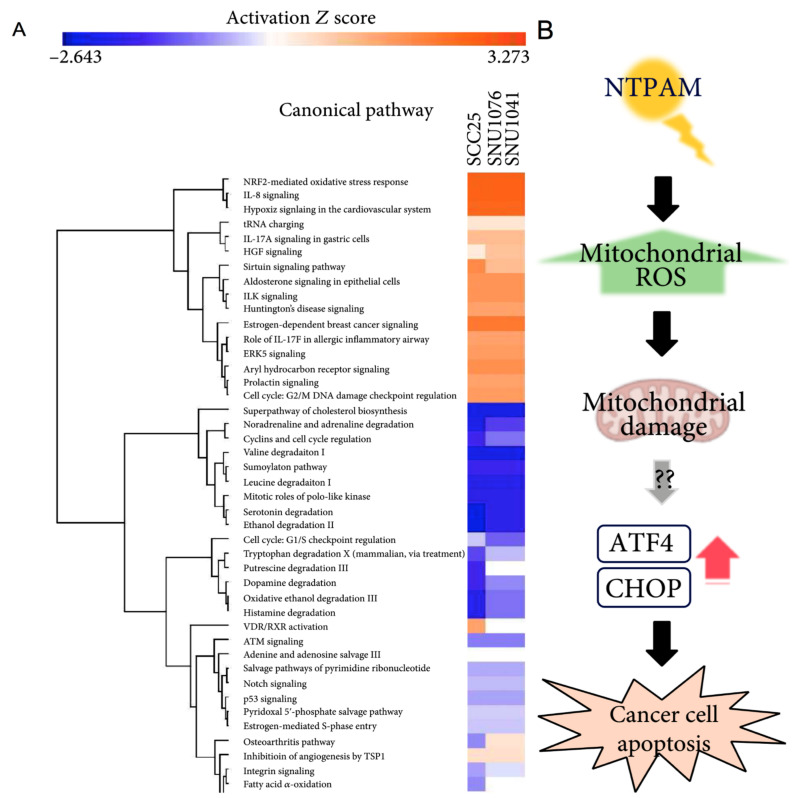
Mechanism and pathways induced by plasma-activated media treatment in head and neck cancer. (**A**) Top relevant canonical pathways associated with plasma activated media treatment by core analysis. (**B**) The proposed mechanism of plasma activated media based anti-cancer effects on head and neck cancer cells [77].

**Figure 6 ijms-23-10238-f006:**
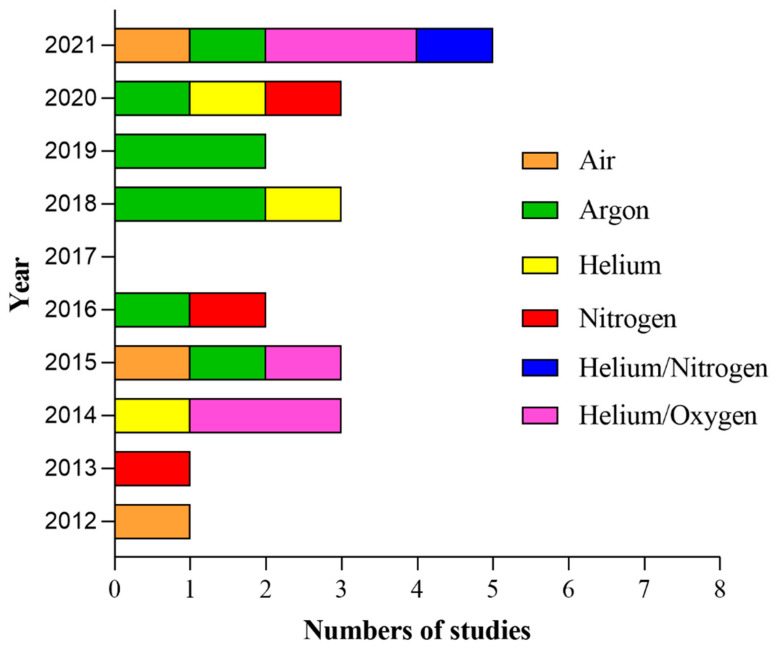
Yearly distribution of the 23 articles (Dai et al. [84] has been excluded because the carrier gas was not specified in the articles) in the present systematic review according to the type of carrier gas.

**Table 1 ijms-23-10238-t001:** List of excluded studies and reasons for exclusion.

Study	Reasons for Exclusion
Furusaka T. et al., 1986 [48]	No plasma source has been used
Akhmedov I.N. et al., 2011 [49]	Article in Russian
Keidar M. et al., 2011 [50]	No head and neck in vitro or in vivo experiments
Virard F. et al., 2015 [51]	No tumoral cell line, but normal primary human fibroblast cultures isolated from oral tissue
Adhikari E.R. et al., 2016 [52]	No head and neck in vitro or in vivo experiments
Laurita R. et al., 2016 [53]	No head and neck cell line, but mesenchymal stem cells derived from human fetal membranes (FM-hMSCs)
Lee J.H. et al., 2016 [54]	No head and neck cell line, but human gingival fibroblasts
Tanaka H. et al., 2016 [66]	U251SP cells (human glioblastoma cell line) and MCF10A cells (human mammary epithelial cell line) as models
Kang S.U. et al., 2017 [55]	No head and neck cell line, but primary fibroblast cell line
Metelmann H.R. et al., 2018 [68]	Case report
Han I. et al., 2018 [56]	Abstract for poster/oral presentation
Yan D.Y. et al., 2018 [63]	No head and neck cell lines have been used in the experiments
Biscop E. et al., 2019 [70]	No head and neck cell lines have been used in the experiments
Ghimire B. et al., 2019 [69]	Agarose has been used as tissue model (no head and neck tissues)
Hajizadeh K. et al., 2019 [57]	Bone marrow mesenchymal stem cells, no head and neck tumoral cell model
Weiss M. et al., 2019 [65]	Primary human fibroblasts isolated from foreskin samples, no head and neck tumoral cell model
Zhang H. et al., 2019 [62]	A549 lung carcinoma cells, no head and neck tumoral cell model
Bekeschus S. et al., 2020 [61]	SK-MEL-28, MeWo, MaMel86a, Panc-1, Miapaca2GR, HeLa, MDA-MB-231, PC-3, 501-Mel, OVCAR3, and A375; no head and neck tumoral cell model
Jaiswal A.S. et al., 2020 [58]	Nasopharyngeal angiofibroma
Jing J. et al., 2020 [59]	LTPA by plasma radiometer
Moritz J. et al., 2020 [67]	SK-Mel 28, MNT-1, Capan-1, PANC-01, HT-29, SW 480, MCF-7, and MDA-MB-231, no head and neck tumoral cell model.
Evert K. et al., 2021 [60]	Only oral normal mucosa model in vivo (mouse)
Wang P. et al., 2021 [64]	LTPA by low-temperature bipolar radiofrequency ablation system

**Table 2 ijms-23-10238-t002:** Characteristics of preclinical studies.

Authors	Level (In Vitro/In Vivo/Ex Vivo)	Models	Cell Type(s)	Plasma Device	Direct/ Indirect Treatment	Condition of Use	Analyses	Results
Choi B.B. et al., 2012 [37]	In vitro	Cell lines	**Normal**: WI-38 **Tumor**: SCC-25	air-NTP	Direct	4 × 10^4^ cells incubated on cover slip (12 mm diameter) for 24 h with the GNP-EGFR, washed with serum free media, and then cover slips placed 2 mm from the plasma source and exposed to treatment. During plasma treatment, cells humidified by 5 μL serum free media.	**Biological**: -Cell viability; -Protein expression; -Cell morphology.	**BIOLOGICAL** **Cell Viability (Trypan blue)**: -Plasma death rate 3.61% in WI-38 and 15.7% in SCC-25; -Plasma plus GNP-EGFR death rate 34% in WI-38 and 92% in SCC-25. **Protein expression (Western blot)**: -Basal EGFR more expressed in SCC-25 cells than in WI-38. **Cell morphology (SEM)**: -SCC-25 morphology, after plasma plus GNP-EGFR changed from spindle to round shape, and cell shrinkage and membrane rupture were observed.
Han X. et al., 2013 [23]	In vitro	Cell line	**Tumor**: SCC-25	N_2_-APPJ	Direct	Before plasma irradiation, SCC-25 were grown on a grid slide with a marked dot at the center placed inside a P35 cultural dish (3 mm thickness). 2.4 mL PBS was added instead of a culture medium to prevent cell desiccation. *Control 1:* treated with N_2_ gas without plasma. *Control 2:* untreated neither plasma nor gas flow.	**Biological**: -Identification and quantification of cells with DNA damage. **Chemical**: -Determination of reactive species.	**BIOLOGICAL** **DNA damage (Immunofluorescence)**: -60% DSB damage after 30 s irradiation; -Nearly 80% at 2 min. **CHEMICAL** **Determination of reactive species (Optical emission spectroscopy)**: -Nitrogen oxide bands (NO-ƴ) detected between 200 and 300 nm; -The bands of the second positive system of molecular nitrogen recorded in the range of 300–420 nm; -The dominant peak around 557 nm represents an excimer of ON_2_ (O(^1^D)N_2_).
Chang J.W. et al., 2014 [36]	In vitro	Cell lines	**Tumor**: MSK QLL1 SCC-1483 SCC-15 SCC-25	He + O_2_^−^ NTP spray type	Direct	Cells were treated in complete medium. *Control:* cells treated with He + O_2_ gas without plasma.	**Biological**: -Apoptosis; -Cell cycle -DNA damage; -Cell morphology; -Protein expression.	**BIOLOGICAL** **Apoptosis and cell morphology (FACS, fluorescence microscopy)**: -Rounding, shrinkage, detachment and increase in annexin V, PI and TUNEL positive cells. **Cell cycle (FACS)**: -Sub-G1 cell cycle arrest in wild-type p53 OSCCs (SCC-25), not in mutated p53; -Decrease in sub-G1 phase cells in the ATM siRNA-transfected compared with the control siRNA-transfected. **DNA damage (COMET assay, immunocytochemistry)**: -Intensity-dependent increase in the number of typical comets with elongated tails; -Increased ƴH2AX foci in the nuclei of SCC-25 cells. **Protein expression (Western blot)**: -NTP increased p-ATM, p-p53, p21, cyclin D1, and ƴH2AX expression in wild-type p53, but not mutated p53 cells.
Guerrero-Preston R. et al., 2014 [5]	In vitro	Cell lines	**Normal**: NOKsi OKF-6 **Tumor**: JHU-022 JHU-028 JHU-029 SCC-25	He CAP	Direct	Cells were treated in complete medium. *Control:* cells treated with He gas without plasma.	**Biological**: -Cell viability; -Colony formation; -Protein expression.	**BIOLOGICAL** **Cell viability (MTT)**: -Dose-response decrease in SCC-25 and JHU-O28; -Decreased in JHU-O22 and JHU-O29 after 30 and 45 s treatment; -Not affected in OKF-2; -SCC-25 and OKF-6 formed colonies after 10 s treatment; -NOKsi cells formed colonies. **Protein expression (Western blot)**: -No occurrence of PARP cleavage.
Kang S.U. et al., 2014 [73]	In vitro	Cell lines	**Tumor**: FaDu HN-9 SNU-1041 SNU-899	He + O_2_-NTP spray type	Direct	3 mL of cell suspension with a concentration of 1 × 10^5^ cells/mL on the petri dish (diameter 60 mm) treated in complete medium (depth 10 mm). *Control:* cells treated with He + O_2_ gas without plasma.	**Biological**: -Apoptosis; -Protein expression; -Mitochondrial damage. **Chemical**: -ROS production.	**BIOLOGICAL** **Apoptosis (FACS, TUNEL)**: -Significant increased apoptosis of treated FaDu, HN-9, SNU-899, and SNU-1041 compared with the control and gas-only groups; -In FaDu, NTP induced apoptosis by MAPK-mediated mitochondrial ROS. **Protein expression (Western blot)**: -Increased expression of p-p38, p-c-JUN N-terminal kinase (JNK), and p-ERK after NTP treatment in FaDu. **Mitochondrial damage (FACS—MMP measurement)**: -Loss of MMP and mitochondrial damage. **CHEMICAL** **ROS production (FACS and confocal microscopy)** -Mitochondrial superoxide levels increase in NTP-treated cells.
Kang S.U. et al., 2014 [73]	In vivo	FaDu-derived xenograft in 16 male BALB/c nu/nu mice.	**Tumor**: FaDu	He + O_2_-NTP spray type	Direct	Daily single treatment, 1 cm apart from the upper margin of tumor for 20 days. *Contro*l: untreated BALB/c nu/nu mice	**Biological**: -Protein expression; -Apoptosis; -Tumor mass measurement.	**BIOLOGICAL** **Protein expression (IHC)**: -Increased caspase-3 and Nox-3 levels. **Apoptosis (TUNEL assay)**: -Increased TUNEL staining, compared with control. **Tumor mass measurement (Sliding caliper)**: -Inhibition of tumor growth after 11 days of treatments.
Kim S.Y. et al., 2015 [74]	In vitro	Cell lines	**Normal**: MRC-5 HNLF **Tumor**: AMC-HN6 Human derived cancer cell lines: FaDu SCC-15 SCC-QLL1 SCC-1483 SNU-1041 Murine derived cancer cell line: SCC-7	LTP produced by He + O_2_-NTP spray type	Indirect	LTP was applied to HNC cells in the absence of serum. *Control Media (CM):* air-treated media.	**Biological**: -Cell viability; -Apoptosis; -Protein expression; -Colony forming assay; -Gene expression; -Detection of AKT ubiquitination. **Chemical**: -ROS production.	**BIOLOGICAL after NTP treatment.** **Cell Viability (MTT on SCC-15 and SCC-QLL1)**: -Significant reduction of HNC viability. **Apoptosis (TUNEL assay on SCC-15 and SCC-QLL1)**: -Apoptotic cell death. **Detection of AKT ubiquitination (SCC-15)**: -AKT degradation via lysine 48-linked ubiquitination. **RT-PCR (SCC-15) and proximity ligation assay (SCC-15, SCC-QLL1)**: -Increased level of MUL1 mRNA; -Cellular ROS increase MUL1/AKT binding and cytotoxicity. **BIOLOGICAL after LTP treatment.** -Decreased SCC-15 viability; -Greater inhibition of colony-forming growth in LTP-treated SCC-15 compared with *CM*; -LTP reduced AKT or p-AKT levels and increased the level of MUL1; -CM did not change the levels of AKT, p-AKT, or MUL1; -LTP reduced AKT and p-AKT levels, which was prevented by MUL1 knockdown. **CHEMICAL** **Measurement of ROS (FACS)**: -NTP induced ROS production in human HNC cells.
Kim S.Y. et al., 2015 [74]	In vivo	SCC-7-derived syngeneic tumor model in 10 female C3H/HeJ mice. SCC-15-derived xenograft in BALB/c nu/nu mice.	**Tumor**: SCC-7 SCC-15	LTP produced by He + O_2_-NTP spray type	Indirect by injection	Daily treatment of 200 μL of medium or LTP for 6 days by intra-tumoral injection.	**Biological**: -IHC; -Protein expression.	**BIOLOGICAL** -Inhibitory effect of tumor development after the fourth treatment and reduction in tumor weight; -Inhibition in tumor volume after the ninth treatment; -Reduction of p-AKT levels and increase in MUL1.
Welz C. et al., 2015 [82]	In vitro	Cell lines	**Tumor**: FaDu OSC-19	air-CAP (SMD)	Direct	Cell culture medium was removed before the CAP treatment and was added immediately after treatment. *Control:* cells not exposed to CAP.	**Biological**: -Cell viability; -DNA damage; -Apoptosis.	**BIOLOGICAL** **Cell viability (MTT)**: -Time-dependent reduction in OSC-19 and FaDu cells compared to control. **DNA damage (COMET assay)**: -Time-dependent DNA fragmentation. **Apoptosis (FACS)**: -No dose-dependent apoptosis in both cell lines.
Lee J.H. et al., 2016 [75]	In vitro	Cell lines	**Normal**: HGF-1 **Tumor**: HSC-2 SCC-15	N_2_-CAP jet	Direct	Cells were treated in complete medium (concentration of 1 × 10^4^ cells/100 μL). *Control:* cells not exposed to CAP.	**Biological**: -Cell viability; -Apoptosis; -Cell cycle; -Protein expression. **Chemical**: -Intracellular ROS detection; -Thiol detection.	**BIOLOGICAL** **Cell Viability (Tetrazolium salt assay)**: -Significant decrease in HSC-2 and SCC-15 compared to HGF-1. **Apoptosis (FACS)**: -More apoptotic death in HSC-2 compared to HGF-1. **Cell Cycle (FACS)**: -HSC-2 killed via sub-G1 arrest; -HGF-1 did not show changes in cell cycle components. **Protein expression (Immunoblot assay)**: -Degradation and dephosphorylation of EGFR in HSC-2 cells; ->deactivation of the EGFR pathway. **CHEMICAL** -ROS measurement by OES.
Chauvin J. et al., 2018 [6]	In vitro	Cell lines	**Tumor**: FaDu monolayer culture FaDu spheroids (MCTS)	PAM produced by He-CAP jet (DBD)	Indirect	PAM was applied to FaDu and FaDu MCTS. *Control:* cells *t*reated with He gas without plasma.	**Biological**: -Cell viability.	**BIOLOGICAL** **Cell viability (PrestoBlue****^®^****)**: -PAM exposure dependent cell detachment from MCTS due to the presence of H_2_O_2_; -A rapid spheroids regrowth due to a resistance of FaDu cells to H_2_O_2_; -Inhibition of cell growth after multiple treatments of MCTS with PAM; -MCTS are brought out when comparing PAM effect on 2D versus MCTS. Inversely, PAM induces cell death in the case of 2D cell culture.
Hasse S. et al., 2019 [72]	In vitro Ex vivo	Cell lines	**Normal**: HaCaT **Tumor**: HNO-97 -10 patients (6 M/4 F); age range: 43–75 years affected by maxillofacial cancer.	PAM produced by Ar-CAP Ar-CAP	Indirect for cell lines. Direct on tissue samples.	PAM was applied to cells. Tissue samples were directly exposed. *Control:* untreated tissue sample and human non-malignant mucosa.	**Biological**: -Cell viability; -Cell cycle; -Protein expression; -Cell motility. -Apoptosis; -DNA fragmentation; -Cytochrome C measurement; -Cytokine detection.	**BIOLOGICAL** **Cell viability (Resazurin-based assay)**: -After plasma treatment, time-dependent reduction in HaCaT and HNO-97. **Cell cycle (FACS)**: -19.3% of HNO-97 cells in G2/M vs. 30% of HaCaT cells. **Global protein expression (Spectrometry)**: -Time dependent activation of caspase 3/7 in both cell types; -Several protein expression changes. **Cell motility (Wound Healing)**: -Reduction in HNO-97; -Normal cells not affected. **Apoptosis/DNA fragmentation (TUNEL assay)**: -Stronger induction of apoptosis in tumor tissue in situ compared to healthy tissue. **Cytochrome C measurement (ELISA)**: -Three-fold increase in tumor tissue. **Cytokine detection (Flow cytometer)**: -IL-22 increased in healthy tissue; -INF-γ TNF-α and IL-10 found predominantly in tumor tissue compared to non-cancerous tissue.
Sato K. et al., 2019 [80]	In vitro	Cell lines	**Normal**: HS-K IMR-90-SV **Tumor**: Ca9-22 Ho-1-u-1 HSC-2 HCS-3 HSC-4 Sa-3 SAS	Ar-NTP	Direct	Cells were treated in complete medium with NTP. *Control:* cells treated with Ar gas without plasma.	**Biological**: -Cell viability; -Apoptosis; -Quantification and localization of catalytic Fe(II); -Migration and Invasion; -Protein expression; -Colony formation. **Chemical** -ROS detection.	**BIOLOGICAL** **Cell viability (WST-8 assay)**: -Time and treatment-dependent decrease in SCC; -Decrease in SAS e CA9-22 viability as the concentration of FAC increased; -Increase as the concentration of DFO increased in co-treatment with NTP. **Quantification and localization of Fe(II) (Flow cytometer)**: -The Fe level in IMR-SV-90 was lower than SAS and CA9-22; -No difference in the Fe levels between SAS and Ca9-22 cells. **Apoptosis (FACS, TUNEL)**: -Apoptotic cells in SAS and treated Ca9-22. **Migration and invasion (Wound Healing and transwell)** -Suppressed migratory and invasive ability of SAS and Ca9-22 compared with control. **Protein expression (Western blot)**: -Decrease in MMP-2 SCC cells compared to control. **Colony formation (Soft agar)**: -Decreased colony formation in treated SAS cells compared with control. **CHEMICAL** **ROS detection (Flow cytometer)**: -Increased ROS levels in treated SCC.
Han X. et al., 2020 [71]	In vitro	Cell lines	**Normal**: OKF-6/T **Tumor**: SCC-25	N_2_-APPJ	Direct	Cells were treated in 2.4 mL of PBS (3 mm depth) with N_2_-APPJ. *Control:* cells treated with N_2_ gas without plasma and no treatment. *Control*: untreated cells	**Biological**: -DNA double strand breaks	**BIOLOGICAL** **DNA double strand breaks (****Immunofluorescence)**: -DNA damage in cancer cells was maximized at the plasma jet treatment region and declined radially outward; -In cancer cells DNA damage decreased slightly over the first 4 h and rapidly decreasing by approximately 60% at 8 h post-treatment; -Damage observed 2 h after treatment in non-malignant cells; -S phase cells were more susceptible to DNA damage than either G1 or G2 phase cells.
Lee C.M. et al., 2020 [38]	In vitro	Cell lines	**Normal**: HGF-1 **Tumor**: SCC-15	Ar-CAP jet	Direct	Cells (1 × 10^5^ cells/100 μL) in medium were treated. Group 1: 1 µM cisplatin + 3 min CAP. Group 2: 3 µM cisplatin + 1 min CAP.	**Biological**: -Cell viability; -Apoptosis; -Protein expression. **Chemical**: -ROS generation; -CAP jet measurement.	**BIOLOGICAL** **Cell Viability (MTT) and Apoptosis (fluorescence microscopy/FACS)**: -Decrease in cell viability by increasing cisplatin concentration and CAP exposure. **Protein expression (Western blot)**: -Increased expression of PTEN and p53 in both cell lines; -Increased expression of cleaved Cas-9 in SCC-15. **CHEMICAL** **ROS generation (Fluorimetric assay)**: -Group 1 and group 2 reported 300% and 500% increase in SCC-15 vs. 130% and 170% in HGF-1. **CAP jet measurement (OES analysis)**: -Peaks of Ar^+^, OH and O^−^ ions observed in the UV range (200 nm–400 nm) and visible range (690 nm–950 nm).
Ramireddy L. et al., 2020 [79]	In vitro	Cell line	**Tumor**: SCC-4	He-CAPP jet	Direct	He-CAPP jet was applied to SCC-4 cells in cell medium. *Control:* cells treated with He gas without plasma.	**Biological**: -Cell viability; -Cell cycle; -Protein expression. **Chemical**: -ROS/RNS detection.	**BIOLOGICAL** **Cell viability (MTT)**: -Significantly greater cell death (dose-dependent) compared to controls. **Cell cycle (FACS)**: -Dose-dependent increase G1 cell cycle. **Protein expression (Western blot)**: -Downregulation of ATM protein, Bax increase, and Bcl-2 decrease compared to controls. **CHEMICAL** **ROS/RNS detection (Fluorimetric assay)**: -He-CAPP time-dependent increase in intracellular ROS and RNS levels; -H_2_O_2_ increased immediately after He plasma treatment but reached basal level after 3 h.
Lin A. et al., 2021 [76]	In vitro	Cell lines	**Tumor**: A-375 CAL-27 U-87	Air-microsecond pulsed DBD	Direct	Cells (15 × 10^4^) were treated with DBD after removing the medium. *Control:* untreated cells.	**Biological**: -Cell viability. **Chemical**: -H_2_O_2_, NO_2_^−^ and NO_3_^−^ measurements	**BIOLOGICAL** **Cell survival (Trypan Blue Assay)**: -Reduced survival for all cell lines; -No significant difference between the different treatment times and pulse frequencies. **CHEMICAL** **H_2_O_2_ (Fluorimetric assay) and NO_2_^−^ /NO_3_^−^ (Colorimetric assay) measurement**: -H_2_O_2_, NO_2_^−^ and NO_3_^−^ concentrations linearly increased with pulse frequency and time of application; -The generation of H_2_O_2_, NO_2_^−^ and NO_3_^−^ depends on the total delivered energy during treatment.
Oh C. et al., 2021 [77]	In vitro	Cell lines	**Normal**: Primary normal human fibroblasts **Tumor**: SCC-25 SNU-1041 SNU-1076	NTPAM produced by He-O_2_-PlasmaJet	Indirect	NTPAM preparation: NTP was exposed to the culture media for various activation times.	**Biological**: -Cell viability; -Apoptosis; -Gene expression; -Protein expression; -Mitochondrial damage; -Cells morphology. **Chemical**: -ROS detection	**BIOLOGICAL** **Cell viability (WST-1)**: -Significantly reduced in all HNC cell lines in a manner dependent on NTP activation time compared to normal control cells. **Apoptosis (FACS)**: -Selectively induced apoptosis in HNC cells. **Gene expression by qRT-PCR analysis**: -NTPAM treatment regulated mechanisms related to cell death and cell cycle; -Significant association with the regulation of PERK, closely related to an altered cell stress response; -ATF4 and CHOP are key regulators of NTPAM induced HNC cell death. **Protein expression (Western blot)**: -Increase in cleaved PARP levels, -Decrease in pro-Caspase 3 and Bcl-2. **Mitochondrial damage (OCR, SEM)**: -Induction of mitochondrial dysfunction in HNC cells through mitochondrial damage. **CHEMICAL** **ROS detection (Fluorescent assay)**: -Significant increase in mitochondrial peroxide levels.
Oh C. et al., 2021 [77]	In vivo	Xenograft Animal Model: 10 healthy male BALB/c nude mice (5 control group, 5 NTPAM treatment group).	**Tumor**: SNU-1041	NTPAM	Indirect	SNU1041 cells (5 × 10^6^ cells/mL) were injected subcutaneously into each mouse. Tumor formation was confirmed on the 10th day after injection. 100 μL intra-tumoral injections of medium in the control group and NTPAM in the experimental group was administered once daily for 11 days.	**Biological**: -Protein expression; -Tumor weight measurement.	**BIOLOGICAL** **Protein expression (Histological and Immunohistochemical analysis)**: -In tumor tissues, low expression levels of Ki-67 and high expression levels of ATF4 and CHOP were observed in the NTPAM-treated group, compared with the normal control group. **Tumor weight measurement**: -After NTPAM treatment, the tumor weight was significantly lower than in control tumors.
Park J. et al., 2021 [78]	In vitro	Cell lines	**Normal**: HaCaT **Tumor**: SCC-25	Ar-NTP	Direct	A day prior, cells were cultivated in a growth medium with or without PD-L1 Ab + GNP. Immediately prior to treatment, the dishes were rinsed with PBS and later positioned under the end of the plasma jet.	**Biological**: -Cell viability; -Apoptosis; -Protein expression.	**BIOLOGICAL** **Cell viability (SRB assay, Fluorescent dyes)**: -The treatment with PD-L1 ab + GNP + NTP significantly increased the number of dead cells in SCC-25 compared to the other treatments. **Apoptosis (Western blot, Immunocytochemistry)**: -Elevated expression levels of cleaved caspase-3 and cleaved PARP in the PD-L1 ab + GNP + NTP group; -AIF and cyt C in the control were clustered in punctuate distribution forms whilst they were disseminated in treated cells.
Sklias K. et al., 2021 [81]	In vitro	Cell lines	**Normal**: 1BR3 hTERT RPE1 (immortalized with hTERT) **Tumor**: CAL-27 FaDu	He + O_2_-DBD micro plasma jet	Indirect Direct	*Indirect:* -PBS in empty well was treated with He/O_2_ PAP. -Tumor or normal cells were then incubated for 1 h with PAP. *Reconstituted buffer (RB)*: PBS adjusted with plasma-induced concentrations of H_2_O_2_, NO_2_^−^, NO_3_^−^ and pH *Direct:* Cell culture medium was removed, and the cells were washed with PBS. Then, PBS was added to cells and exposed to plasma for 1 h.	**Biological**: -Cell viability; -Apoptosis. **Chemical**: -ROS detection; -Lipid peroxidation measurement.	**INDIRECT TREATMENT** **Cell viability (MTT, trypan-blue and CellTiterGlo^®^)**: -Lower in FaDu compared to CAL-27; -The strongest cytotoxic effect at a higher concentration of RONS and acidic pH. **Apoptosis: Caspase-Glo^®^ 3/7 Assay System.** **ROS detection (OES)**: -The highest concentration of H_2_O_2_, NO_2_^−^, NO_3_^−^ is obtained after 12 min plasma treatment, at a gas flow of 0.5 slm and at a treatment distance of 8 mm; -These conditions induced the strongest reduction in pH; -H_2_O_2_ is a master player in PAP-induced cancer cell death since the addition of catalase during PAP treatment prevents the toxicity of PAP. **RECONSTITUTED BUFFER** RB is as efficient as PAP to induce lipid peroxidation, intracellular ROS formation, caspase 3/7 activity and cell death in FaDu and CAL-27 cell lines. **DIRECT TREATMENT OF CAL27 AND FADU CELL LINES** -Greater CAL-27 and FaDu death in direct than indirect plasma treatment **DIRECT TREATMENT OF NORMAL CELLS** Strong reduction of 1Br3 and RPE-hTERT metabolic activity of normal cell lines direct plasma treatment, while no change after indirect and reconstituted buffer from 6 h up to 72 h post treatment.
Wu C.Y. et al., 2021 [83]	*In vitro*	Cell lines	**Tumor**: SAS CAL-27 FaDu Detroit 562	N_2_+He-NT micro plasma jet	Direct	After cells exposure to plasma, the medium was changed with a new fresh one and incubated for further 24 h.	**Biological**: -Cell viability; -Apoptosis. **Chemical**: -ROS and RNS detection	**BIOLOGICAL** **Cell viability (MTS)**: -Significant decrease with plasma exposure time in SAS, CAL-27 and FaDu; -Significant less decrease in Detroit 562 cells between 30 and 60 s. **Apoptosis (FACS/TUNEL)**: -After plasma exposure for 5 min, 7.8-fold increase in the apoptotic percentage compared to non-treatment group. **CHEMICAL** **ROS and RNS detection (OxiSelect Intracellular Assay kits)**: -ROS and RNS concentration in medium increased with plasma exposure time.

**1BR-3**: human normal skin; **A-375**: human melanoma; **Ab**: antibody; **AIF**: apoptosis-inducing factor; **AMC-HN6**: human floor of the mouth squamous cell carcinoma; **AMC-HN-9**: human undifferentiated carcinoma of the parotid gland; **APPJ**: atmospheric pressure plasma jet; **Ar**: argon; **ATF4**: activating transcription factor 4; **ATM**: ataxia telangiectasia mutated; **BAX**: BCL2 associated X; **BCL-2**: B-cell lymphoma-2; **BR3**: human primary skin fibroblasts; Ca9-22: Human gingival squamous cell carcinoma; **CAL****-27**: human tongue squamous cell carcinoma; **CAP**: cold atmospheric plasma; **CAPP**: cold atmospheric pressure plasma; **Cas-9**: caspase-9; **CHOP**: C/EBP homologous protein; **cyt C**: cytochrome C; **DBD**: dielectric barrier discharge; **DFO**: desferrioxamine; **Detroit-562**: human pharyngeal squamous cell carcinoma; **DSB**: DNA double-strand break; **ep**: energy per pulse; **FAC**: ferric ammonium citrate; **FACS**: fluorescence activated cell sorting; **FaDu**: human hypopharynx squamous cell carcinoma; **Fe**: iron; **GNP**: gold nanoparticles; **GNP-EGFR**: anti-epidermal growth factor (**EGFR**) antibody conjugated gold nanoparticle (**GNP**); **GNP**: gold nanoparticles; **HaCaT**: human nonmalignant keratinocytes; **He**: helium; **HGF-1**: human gingival fibroblasts; **HNLF**: human normal lung fibroblast; **HN-9**: human tongue squamous cell carcinoma; **HNO-97**: human tongue squamous cell carcinoma; **HNC**: squamous cell carcinoma of the head and neck; **Ho-1-u-1**: human floor of the mouth squamous cell carcinoma; **HSC-2**: human oral cavity squamous cell carcinoma; **HSC-3**: human tongue squamous cell carcinoma; **HSC-4**: human tongue squamous cell carcinoma; **HS-K**: human kidney fibroblasts; **hTERT RPE1**: human retinal pigment epithelial cells; **IHC**: immunohistochemistry; **JHU-022**: human laryngeal squamous cell carcinoma; **JHU-028**: human lung adenocarcinoma; **JHU-029**: human laryngeal squamous cell carcinoma; **ICC**: immunocytochemistry; **IMR-90-SV**: human lung fibroblasts; **LTP**: liquid-type NTP; **MMP**: mitochondrial membrane potential; **MRC-5**: human lung fibroblast; **MSK-QLL1**: human head and neck squamous cell carcinoma; **MCTS**: multi cellular tumor spheroids; **N**: nitrogen; **N/A**: not applicable; **NOKsi**: human normal oral keratinocytes; **NTP**: non-thermal atmospheric pressure plasma; **NTPAM**: non-thermal atmospheric plasma-activated media; **O_2_**: oxygen; **OCR**: oxygen consumption rate; **OES**: optical emission spectroscopy; **OKF6/T**: human normal floor of the mouth keratinocytes; **OSC-19**: human low graded tongue squamous cell carcinoma; **PAM**: plasma activated medium; **PAP**: plasma activated PBS; **PARP**: poly adenosine diphosphate-ribose polymerase; **PBS**: phosphate buffered saline; **PD-L1**: programmed death-ligand 1; **PERK**: protein kinase R-like endoplasmic reticulum kinase; **PI**: propidium iodide; **PTEN**: phosphatase and tensin homolog; **RNS**: reactive nitrogen species; **ROS**: reactive oxygen species; **SAS**: human tongue squamous cell carcinoma; **SCC-1483**: human oral retromolar trigone cavity squamous cell carcinoma; **SCC-15**: human tongue squamous cell carcinoma; **SCC-25**: human tongue squamous cell carcinoma; **SCC-4**: human tongue squamous cell carcinoma; **SCC-7**: murine squamous carcinoma cells; **SCC-QLL1**: human oral cavity cancer; **SEM**: scanning electron microscope; **SMD**: surface micro discharge; **SNU-1041**: human hypopharynx squamous cell carcinoma; **SNU-1076**: human laryngeal squamous cell carcinoma; **SNU-899**: human laryngeal squamous cell carcinoma; **SRB**: sulforhodamine B; **td**: treatment distance; **tt**: treatment time; **TUNEL**: terminal deoxynucleotidyl transferase dUTP nick end labeling; **U-87**: human glioblastoma; **WB**: western Blotting; **WI-38**: human normal lung fibroblasts; **WST-1**: water-soluble tetrazolium salt-1.

**Table 3 ijms-23-10238-t003:** Characteristics of clinical studies.

Authors	Tumor Type(s)	Sample Size	Numbers and Characteristics of Groups	Stage	Plasma Device	Direct/Indirect Treatment	Primary OUTCOMES	Secondary Outcomes	Results	Follow-Up
Metelmann H.R. et al., 2015 [85]	Advanced HNC	12	-Female: 6 -Male: 6 -Caucasian -Age range: 50–77 years. -Karnofsky performance status: 60–80.	T 4 N 0–3 M 0–1	Ar-CAP kINPen MED.	Direct: spot exposure of the ulceration to CAP.	-Change in contamination; -Patient-expressed need of pain medication; -Side effects.	Anti-cancer effects (tumor growth).	-Relief and reduction of fetid odor and pain; -Bacteria decrease in cancer ulcerations; -not severe side effects; -Visible effects after 2 weeks of exposure on tumor surface.	N/P
Schuster M. et al., 2016 [87]	Advanced HNC	21	-Female: 9 -Male: 12 -Caucasian -Age range: 40–77 years. -Karnofsky performance status: 60–80. *Group I:* (*n* = 12) palliative CAP treatment *Group II:* (*n* = 9) curative CAP + surgery treatment.	N/P	Ar-CAP kINPen MED.	Direct: spot exposure of the ulceration to CAP.	Tumor surface responses. *Type 1:* flat area with vascular stimulation; *Type 2:* contraction of tumor ulceration rims with scabs surrounded by tumor tissue in visible progress.	Evaluation of CAP-induced visible tumor surface in relation to CAP-induced apoptotic cell kill.	-No sign of enhanced or stimulated tumor growth in any patient; -More apoptotic cells in tissue areas treated with CAP than in untreated; -Visible tumor surface response in relation to apoptotic cell.	Tumor surface response evaluated by photographic analysis after 2 weeks of treatments.
Metelmann H.R. et al., 2018 [15]	Advanced OPSCC	6	-Female: 3 -Male: 3 -Age range: 53–78 years	Locally advanced cancer of the oropharynx (pT4) with contaminated tumor ulcerations.	Ar-CAP kINPen MED.	Direct: spot exposure of the ulceration to CAP.	-Survival time; -Course of disease; -Tumor remission; -Safety of treatment.	Incisional biopsies were performed to verify changes at the cellular level.	-Tumor reduction; -Significant improvement in tumor decontamination (reduction of odor) and tumor mass; -Palliation in terms of quality of life.	Related to the death of participant.
Schuster M. et al., 2018 [86]	Advanced HNC	20	-Female: 10 -Male: 10 -Caucasian -Age range: 49–84 years -Karnofsky performance status: 60–80.	N/P	Ar-CAP kINPen MED.	Direct: spot exposure of the ulceration to CAP e.	-Evaluation general health condition and side effects; -History of cancer treatment; -State of disease; -Palliation procedure.	N/P	Side effects were mild to moderate and never life threatening.	N/P
Dai X. et al., 2020 [84]	LC	100	-Female: 42 -Male: 28 -Age range: 60–85 years old *Control:* 50 treated with conventional surgery *Test:* 50 treated with CAP	N/P	Unitec low-temperature plasma operation system.	Tumor was ablated with a low-temperature plasma cutter with the extent expanded to 3–5 mm away from the edge of the lesion.	-Postoperative efficacy; -Influence on the tumor markers, COX-2, and VEGF expressions.	-Operation time -VAS pain; -Mucosal recovery scores.	-Few postoperative complications; -Decreased expression levels of postoperative tumor markers (COX-2 and VEGF).	N/P

**CAP**: cold atmospheric plasma; **COX-2**: cyclooxygenase-2; **HNC**: squamous cell carcinoma of the head and neck; **LC**: laryngeal carcinoma; **N/P**: not provided; **OPSCC**: oropharyngeal squamous cell carcinoma; **VAS**: visual analog scale; **VEGF**: vascular endothelial growth factor.

**Table 4 ijms-23-10238-t004:** Characteristics of plasma devices.

Authors	Plasma Device Description	Pulse Frequency	Pulse	Flow Rate	Gas	Plasma Temperature	Application Distance	Application Time	Total Energy	Power	Manufacturer
Choi B.B. et al., 2012 [37]	air-NTP: the size of the device is 10.24 cm^2^. The mask pattern was etched by wet etching technique on the Cu electrodes which is surrounding both side of PTFE dielectric surface.	22 kHz	15 kV	N/P	Air	N/P	2 mm	30 s	9.2 J/cm^2^	3.15 J/s	Pohang University of Science and Technology, Pohang, Korea (Kim et al., 2010 [88])
Han X. et al., 2013 [23]	N-APPJ: Two copper ribbon electrodes of 0.6 mm thick separated by 1.8 mm are wrapping around a quartz tube (outer diameter: 3 mm). One helical electrode is connected to a HV power supply, and the other is grounded. The electrode wrapping zone has a vertical length of 35 mm.	28 kHz	22.4 kV (Vrms. 7.75 kV) 59 mA (Irms. 17 mA)	1.5 L/min	N_2_	N/P	20 mm	10, 30, 60, and 120 s	N/P	N/P	N/P
Chang J.W. et al., 2014 [36]	He+O-NTP spray type: arc-free and antistatic plate to provide a uniform plasma jet.	20–30 kHz	2–13 kV	N/P	He/O_2_	N/P	N/P	1 s	N/P	N/P	Pohang University of Science and Technology, Pohang, Korea
Guerrero-Preston R. et al., 2014 [5]	He-CAP: The CAP device contains 4 blocks. Block 1 is a DC power supply. Block 2 is a centrally powered electrode with a ground outer electrode wrapped around a quartz tube. Block 3 consists of a capacitor, a transistor, and a timer; block 4 is the He gas supply.	N/P	8 kV	10 L/min^−1^ test and 20 L/min^−1^ positive treatment control	He	N/P	3 cm	10, 30 45 s and 10 s for control	N/P	N/P	School of Engineering and Applied Science of The George Washington University
Kang S.U. et al., 2014 [73]	He + O NTP spray-type: NTP with a newly designed arc-free and antistatic plate to provide uniform NTP for biological research applications. The plasma source is equipped with a pair of electrodes that is made of Al_2_O_3_ (high voltage and ground electrodes, 1040 mm^2^ dimension, 2 mm gap between electrodes) that is isolated from direct contact with the plasma by a ceramic barrier.	20–30 kHz	2–13 kV	N/P	He/O_2_	35 °C	In vitro: 3 cm In vivo: 1 cm	In vitro: N/P In vivo: 20 s	N/P	N/P	N/P
Kim S.Y. et al., 2015 [74]	LTP produced by He + O-NTP spray-type: newly designed arc-free and antistatic plate to provide uniform NTP. The plasma source is equipped with a pair of electrodes made of Al_2_O_3_ (high-voltage and ground electrodes, 10 × 40 mm^2^ in dimension, 2 mm gap between electrodes) isolated from direct contact with the plasma using a ceramic barrier.	20–30 kHz	2–13 kV	N/P	He/O_2_	35 °C	1–2 cm from the culture media	in vitro: 15 min. syngenic tumor model: 1 week. xenograft model: 10 times.	N/P	N/P	N/P
Metelmann H.R. et al., 2015 [85]	kINPen MED: Hand-held unit discharges plasma under atmospheric conditions, requiring a DC power unit and Ar gas reservoir. In the center of a ceramic capillary (inner diameter 1.6 mm) a pin -type electrode (1 mm diameter) is mounted. The needle is powered by a miniaturized RF generator.	1 MHz, modulated with 2.5 kHz and plasma duty cycle of 1:1.	Sinusoidal voltage waveforms range from 2 kV to 3 kV amplitude peak.	5 slm	Ar	38 °C	Spot exposure of the ulceration to CAP from 8 mm.	Cycles of 3 single treatments within 1 week, followed by an intermittence of 1 week without CAP exposure. Repeatedly scanning tumor ulceration accessible area for 1 min/cm^2^. Total treatment time increase to more than 30 min for patients with large ulceration exceeding 30 cm^2^.	N/P	N/P	Neoplas tools GmbH, Greifswald, Germany
Welz C. et al., 2015 [82]	air-CAP: MiniFlatPlaSter is equipped with a high voltage power supply, accumulators, and a SMD electrode for production in air. The SMD electrode consists of a copper foil layer (around 0.2 mm thick), an Epoxy board (1 mm thick), and a stainless-steel mesh of 28 mm in diameter, so that it exactly fits the rim of one well.	6.75 kHz	7 kV	N/P	Air	N/P	17.5 ± 0.5 mm	30, 60, 90, 120 and 180 s	N/P	N/P	FlatPlaSter, Regensburg University Hospital
Lee J.H. et al., 2016 [75]	N-CAP jet: made up of an inner electrode made of tungsten with 1.2 mm depth and 0.2 mm thickness with 3.2 mm depth of quartz as a dielectric. The hole in the outer electrode made of stainless steel was 0.7 mm via the 2 mm height of porous alumina having a 150~200 μm pore size with 35% porosity.	60 Hz	1.2 kV	250; 500; 750; 1000; 1500; 2000 sccm	N_2_	N/P	8 mm	1 min	0.51; 0.62; 1.98; 2.91; 2.4; 2.33 W	N/P	Kwangwoon University
Schuster M. et al., 2016 [87]	kINPen MED: Hand-held unit discharges plasma under atmospheric conditions, requiring a DC power unit and Ar gas reservoir. In the center of a ceramic capillary (inner diameter 1.6 mm) a pin -type electrode (1 mm diameter) is mounted. The needle is powered by a miniaturized RF generator.	1 MHz, modulated with 2.5 kHz and plasma duty cycle of 1:1. Amplitude peaks at a frequency of 1 MHz and modulated with 2.5 kHz.	Sinusoidal voltage waveforms range from 2 kV to 3 kV amplitude peak.	N/P	Ar	N/P	Spot exposure of the ulceration to CAP from a distance of 8 mm	Group 1: cycle of 3 single treatments within 1 week for 1 min, followed by an intermittence of 1 week without CAP exposure. Group 2: one-time application for 3 min followed by total resection of the tumor.	N/P	N/P	Neoplas tools GmbH, Greifswald, Germany.
Metelmann H.R. et al., 2018 [15]	kINPen MED: Hand-held unit discharges plasma under atmospheric conditions, requiring a DC power unit and Argon gas reservoir. In the center of a ceramic capillary (inner diameter 1.6 mm) a pin -type electrode (1 mm diameter) is mounted. The needle is powered by a miniaturized RF generator.	1 MHz, modulated with 2.5 kHz and plasma duty cycle of 1:1.	Sinusoidal voltage waveforms range from 2 kV to 3 kV amplitude peak.	5 slm.	Ar	38 °C	8 mm, vertically to naturally moist tissue surface.	Cycles of 3 single treatments within 1 week, followed by an intermittence of 1 week without CAP exposure. Repeatedly scanning tumor ulceration accessible area for 1 min/cm^2^. Total treatment time increase to more than 30 min for patients with large ulceration exceeding 30 cm^2^.	N/P	N/P	Neoplas tools GmbH, Greifswald, Germany.
Chauvin J. et al., 2018 [6]	PAM produced by He-CAP jet (DBD): made up of Al tape electrodes wrapped on a quartz tube with small diameters (2 mm inner diameter and 4 mm outer diameter) separated by 10 mm space.	10 kHz	10 kV square pulses	3 L/min	He	N/P	2 cm	0, 30, 60, 120, 240 s	N/P	N/P	Université de Toulouse-LAPLACE
Schuster M. et al., 2018 [86]	kINPen MED: Hand-held unit discharges plasma under atmospheric conditions, requiring a DC power unit and Ar gas reservoir. In the center of a ceramic capillary (inner diameter 1.6 mm) a pin -type electrode (1 mm diameter) is mounted. The needle is powered by a miniaturized RF generator.	1 MHz, modulated with 2.5 kHz and plasma duty cycle of 1:1. Amplitude peaks at a frequency of 1 MHz and modulated with 2.5 kHz.	Sinusoidal voltage waveforms range from 2 kV to 3 kV amplitude peak	N/P	Ar	N/P	Spot exposure of the ulceration to CAP from a distance of 8 mm.	Cycles of 3 single treatments within 1 week, followed by an intermittence of 1 week without CAP exposure, exceptionally due to the patient’s individual circumstances of 2 to 3 weeks. Repeatedly scanning tumor ulceration accessible area for 1 min/cm^2^.	N/P	N/P	Neoplas tools GmbH, Greifswald, Germany.
Hasse S. et al., 2019 [72]	PAM produced by Ar-CAP kINPen MED made of two electrodes, a pin type high voltage electrode inside a ceramic capillary and one grounded electrode. The plasma is generated at the tip of the pin type electrode and expanded about 1 cm to the surrounding air outside the capillary.	It generates a radiofrequency signal of about 1 MHz. The discharge is switched on at a frequency of 2.5 kHz (50:50).	2–3 kV	5 slm	Ar	35–39 °C	Medium indirect and 8 mm from tissue	In vitro: 20, 40, 80, 150 s Ex vivo: 3 min	N/P	N/P	Neoplas tools, Greifswald, Germany
Sato K. et al., 2019 [80]	Ar-NTP: 2 electrodes 20 mm apart. NTP had an ultrahigh electron density and an O density of approximately 4 × 10^15^ cm^3^.	60 Hz	10 kV	2 L/min	Ar	25 °C	8 mm	30–120 s	N/P	N/P	Habahiro instrument from Prof. M. Hori, Plasma Nanotechnology Research Center, Nagoya University, Japan.
Dai X. et al., 2020 [84]	N/P	N/P	N/P	N/P	N/P	N/P	3–5 mm	N/P	N/P	N/P	Unitec low-temperature plasma.
Han X. et al., 2020 [71]	N-APPJ: 2 copper ribbon electrodes of 0.6 mm thick separated by a distance of 1.8 mm are spirally and alternatively wrapping around a quartz tube, whose outer diameter is 3 mm. One of the helical electrodes is connected to a HV power supply and the other is grounded. The electrode wrapping zone, with a vertical length of 35 mm along the quartz tube, is the major region of plasma ignition. A wider glass tube is sealed outside the quartz tube and a fluid with a high dielectric constant was filled within the volume between these 2 tubes. When N_2_ is introduced into the quartz tube and HV is applied, plasma is ignited and forms a plasma jet of a few cm long to the open atmosphere.	28 kHz	22.4 kV	1.5 slm	N_2_	N/P	20 mm	120 s	N/P	N/P	N/P
Lee C.M. et al., 2020 [38]	Ar-CAP jet: CAP apparatus P500-SM consists of a gas supply system, MFC, a plasma jet and a high-voltage AC power supply.	20 kHz	8.5 kV	5 slm	Ar	N/P	3 cm	10 s–5 min	45 W	N/P	Sakikake Co. Ltd., Kyoto, Japan
Ramireddy L. et al., 2020 [79]	He-CAP jet: Dielectric barrier and quartz tube inner and outer diameters of 2 and 4 mm respectively. Two electrodes copper strips (grounded 2.5 cm and powered 1.5 cm) wrapped around the quartz tube. The powered electrode is 2 mm from the nozzle end and the distance between the grounded and powered electrodes is 1.5 cm.	10 kHz	7.5 kV	5 slm	He	N/P	3 cm	1, 3, 5 min	N/P	N/P	N/P
Lin A. et al., 2021 [76]	air-microsecond-pulsed DBD: plasma system (custom built): Copper electrode covered with 0.5 mm fused silica diameter 1.2 cm	0.05 kHz a 0.5 kHz.	30 kV	N/P	Air	N/P	1–10 mm	10–240 s	9.4 J	N/P	The power supply was custom built (Megaimpulse Ltd.)
Oh C. et al., 2021 [77]	NTPAM generated by NTP jet. The device is composed of a quartz tube (diameter: outer 6 mm, inner 4 mm) with two electrodes (an inner stainless-steel tube and an outer ground ring). The inner is also placed as a gas inlet.	20 kHz	Few kV	He: 4 slpm; O_2_: 1 sccm.	He/O_2_	N/P	In vitro: approximately 1 cm. In vivo: N/A	In vitro: various activation times. In vivo: 100 μL intratumoral injections of NTPAM in the experimental group administered once daily for 11 days.	Discharge power varied from 10 to 24 W.	N/P	N/P
Park J. et al., 2021 [78]	Ar-NTP: 1 dielectric and 2 electrodes. The inner is composed of stainless, the external is wrapped with copper tape. The NTP is produced between the inner and outer electrodes, and the plasma flow temperature is kept below 35 °C for 10 min.	N/P	N/P	2 slm	Ar	N/P	10 mm	5 min	N/P	N/P	Feagle Company (Yangsan-si, Kyeongsangnam-do, Korea)
Sklias K. et al., 2021 [81]	He+O-DBD micro-plasma jet: Plasma reactor: stainless-steel needle (0.7 mm inner and 1.4 mm outer diameters), inserted inside a dielectric tube made of quartz, and biased electrically by applying high voltage square positive pulses. The distance between the needle’s tip and the reactor’s nozzle is fixed (55 mm). The ground electrode (10 mm width), made of copper, is wrapped around the dielectric tube, centered at the tip of the needle.	10 kHz	Amplitude of 6 kV, pulse width of 4.8 µs, rise and fall times of around 25 ns.	0.5 slm or 1 slm	He/O_2_	23 °C	8 or 20 mm.	N/P	N/P	N/P	N/P
Wu C.Y. et al., 2021 [83]	N-NTP jet: Custom-made micro-plasma jet source. One capillary electrode is the jet source to inject additive N_2_. A stainless-steel capillary tube (diameter of 0.8 mm) at the center of the quartz tube is used as the inner electrode. This is connected to the ground and used as a N_2_ flow channel. A piece of copper is used as the outer electrode, which is connected to the output of the generator.	13.56 MHz	12–14 W	N/P	He/N_2_	37 °C	4 mm	-30, 90, 120 s. -5 min (apoptosis).	N/P	N/P	RF, ENI ACG-3B, MSK Instruments, Inc., USA

**APPJ**: atmospheric pressure plasma jet; **Ar**: Argon; **CAP**: cold atmospheric plasma; **Cu**: copper; **DBD**: dielectric barrier discharge; **DC**: direct current; **ep**: energy per pulse; **He**: helium; **HV**: high voltage; **LTP**: liquid-type NTP; **MCF**: mass flow controller; **N**: nitrogen; **N/P**: not provided; **NTP**: non-thermal atmospheric pressure plasma; **O_2_**: oxygen; **PAM**: plasma activated medium; **PAP**: Plasma activated PBS; **PBS**: Phosphate buffered saline; **PTFE**: polytetrafluoroethylene; **RF**: radio frequency; **sccm**: standard cubic centimeters per minute; **slm**: standard Liter/min; **SMD**: surface mounted device; **td**: treatment distance; **tt**: treatment time.

## Data Availability

Not applicable.

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
