# Peer review of "Open Questions in Cold Atmospheric Plasma Treatment in Head and Neck Cancer: A Systematic Review"

_ijms, 2022, doi:10.3390/ijms231810238_

Round 1

Reviewer 1 Report

There are some minor spelling error (line 112), missing dehinite article (line 16).

This paper is clear, well written and informative for the readers. However, I wish some comments or suggestions for standardization of plasma sources for HNC therapy to be included in Conclusion section.

Reviewer 2 Report

The paper is a systematic review of some aspects of head and neck cancer treatment with cold atmospheric plasma.

The topic is important given the growing interest in the biological effects of cold atmospheric plasma and its possible use in medicine.

However, there are some weaknesses that the authors must correct:

1. Authors should upgrade to the PRISMA 2020 guidelines for systematic reviews, rather than the 2010 ones.

2. The authors mis the register name and the registration number at PROSPERO database.

3. Authors must update the flow diagram, according to PRISMA 2020 guidelines.

4. Authors must update the survey date.

5. The search formulas for the various databases used must be shown.

6. The formulas shown does not have MESH terms or quotation marks in the compound terms, so there is no guarantee that all papers have been found.

7. Authors must confirm reference 43 indicated in point 4.6 (Risk and Bias assessment).

8. References must appear in accordance with the journal's rules, harmonized, as well as the way of writing the DOI.
